# Strong and Lightweight Stereolithographically 3D-Printed Polymer Nanocomposites with Low Friction and High Toughness

**DOI:** 10.3390/polym14173628

**Published:** 2022-09-02

**Authors:** Manuel Alejandro Ávila-López, José Bonilla-Cruz, Juan Méndez-Nonell, Tania Ernestina Lara-Ceniceros

**Affiliations:** 1Advanced Functional Materials & Nanotechnology Group, Nano and Micro Additive Manufacturing of Polymers and Composite Materials Laboratory “3D LAB”, Centro de Investigación en Materiales Avanzados S. C. (CIMAV-Subsede Monterrey), Av. Alianza Norte 202, Autopista Monterrey-Aeropuerto Km 10, PIIT, Apodaca C.P. 66628, Nuevo León, Mexico; 2Cinvestav Unidad Saltillo, Calle Industria Metalúrgica No. 1062, Ramos Arizpe C.P. 25900, Coahuila, Mexico

**Keywords:** three-dimensional (3D) printing, TiO_2_, polymer nanocomposites, vat polymerization, SLA, toughness

## Abstract

Strong and lightweight polymer nanocomposites with low friction, high toughness, and complex shapes were obtained for the first time through an affordable stereolithographic 3D printer, using low amounts of TiO_2_ nanoparticles. Tridimensional solid structures (i.e., tensile bars, compressive test specimens, gyroid-type structures, and dense lattices) were obtained. Herein, we found that the compressive stress, compressive strain, yield strength, and toughness corresponding to 3D-printed polymer nanocomposites were simultaneously increased—which is uncommon—using low amounts (0.4 wt.%) of TiO_2_ nanoparticles. Furthermore, we obtained lightweight cylindrical structures exhibiting high resistance to compression with a low friction coefficient (µ~0.2), and the printability of complex and hollow structures was demonstrated.

## 1. Introduction

Today, nanofillers are in high demand to produce strong, high-toughness, and lightweight polymer nanocomposites with complex geometries using low nanofiller loads (<1 wt.%). Specifically, achieving this kind of composite is crucial for the aerospace and automotive industries. Aircraft with lighter components can save up to half a ton in weight, leading to reduced fuel consumption and outstanding decreases in CO_2_ emissions to the atmosphere every year. [1] In this sense, polymer nanocomposites with complex shapes can be obtained via additive manufacturing (AM), also known as 3D printing. [2] This innovative technology allows the production of 3D objects by placing thin layers, one on top of the other (layer-by-layer), following a customizable computer-aided design (CAD), with cost-effective fabrication and high repeatability.

Stereolithography (SLA) is a vat polymerization technique that produces high-resolution 3D objects with excellent fidelity among all AM technologies. A photoresin (i.e., a mixture of monomers, a crosslinker agent, a light-activated photoinitiator, and prepolymers) is selectively photopolymerized by a laser in a vat. [3] SLA enables the production of hollow composite materials, lightweight structures, and components, helping to decrease the so-called “energy burden” derived from fuel consumption. Furthermore, slurries or suspensions of nanofillers such as aluminum nitride [4], γ-Al_2_O_3_ [5], TiO_2_ anatase [6], Ag NPs [7], and organoclays [8] in photoresins have been used recently [9] to enhance the thermal and photonic behavior [10,11], increase the bioactivity in implants [12,13], and increase the electrical conductivity or mechanical properties [2,14] of the resulting composites. Nonetheless, high amounts of nanofillers (1–50 wt.%) have been used to this end.

It is worth highlighting that scientific studies regarding 3D-printed polymer nanocomposites produced by SLA based on photoresin and TiO_2_ nanoparticles (TiO_2_ NPs) are still uncommon. In 2020, Mubarak et al. [15], analyzed 3D-printed polymer composites based on an acrylate-based UV curable resin, using 1% *w*/*w* TiO_2_ NPs under different annealing conditions. The authors found that annealing temperatures of 800 °C promoted 3D-printed polymer composites with the highest thermal, mechanical, and tensile properties. Moreover, Kozlov et al. [16], prepared 3D-printed samples by digital light processing (DLP-SLA) using slurries with high amounts of TiO_2_ NPs (amorphous, 20–50 wt.%; or crystalline, 12–30 wt.%) and 1 wt.% BYK-985 as a dispersing agent. Recently, in 2021, Aktitiz et al. [17] analyzed the mechanical reinforcement of 3D-printed polymer structures by SLA using slurries with 0.25, 0.5, and 1 wt.% TiO_2_ particles (particle size below 1 µm). The authors observed micropores and particle agglomerations within the polymers. Moreover, the brittleness level was increased with the amount of TiO_2_, without any relevant changes in the thermal properties. Therefore, more comprehensive scientific studies regarding 3D-printed polymer nanocomposites produced by SLA based on photocurable resin/TiO_2_ NPs should be developed to achieve more in-depth knowledge about (i) the underlying mechanisms between the polymer matrix and the TiO_2_ NPs during the photopolymerization by SLA; (ii) the role of the low amount of nanofillers (<1 wt.%) in achieving strong and lightweight polymer nanocomposites, as well as the study of their potential applications; and (iii) the printability of complex and hollow structures analyzing their surface and surface finishing.

Herein, solid and lightweight 3D-printed polymer nanocomposites with a low friction coefficient and high toughness were produced by SLA for the first time using very low amounts of TiO_2_ NPs. As a result, the compressive stress, strain, yield strength, and toughness increased simultaneously. Moreover, lightweight (3 g), strong, and highly compression-resistant (5 tons, increasing by around 200%) solid cylindrical structures with a low friction coefficient (µ~0.2) were obtained using only 0.4 wt.% TiO_2_ NPs. In addition, we demonstrated the ability to print complex and hollow nanocomposite structures.

## 2. Materials and Methods

Gray resin (V4—a blend of proprietary methacrylate monomers, oligomers, and photoinitiator from Formlabs^®^ Boston, MA-USA), TiO_2_ NPs (P25, Aeroxide^®^ 99.5%, from Evonik-Germany), and 1-propanol (IPA, CTR ≥ 99.7%) were obtained.

**3D-printed nanocomposites by SLA:** Nanocomposites based on resin/TiO_2_ NPs (R_TiO_2_ X wt.%; where X = 0.0, 0.2, 0.4, 0.8, or 1.2, and represents the *w*/*w*% of the nanoparticles) were obtained following a straightforward methodology. In the first step, the appropriate amount of TiO_2_ NPs (wt.% based on 130 g of resin) was dispersed in 3 mL of IPA using a tip sonicator for 10 min (Cole-Parmer ultrasonic processor) with an amplitude of 50%, a pulse interval of 10 s, and a pulse time of 1 min. Then, the resulting dispersion was added to 130 g of Formlabs gray resin and mixed by mechanical stirring. Six ASTM D638 Type IV tensile specimens for each dispersion were 3D printed at an orientation angle of 45°; meanwhile, six ASTM D695 compression test specimens, six hollow structures (gyroid type), and six complex structures (dense lattices) were 3D printed in a horizontal orientation, as shown in Figure 1. The tridimensional structures were obtained using an affordable and commercial Formlabs-Form 2^®^ 3D printer desktop stereolithography, using a 405 nm violet laser (Class 1), layer thickness of 100 μm, power of 250 mW, and a laser spot size of 140 μm. The printed objects were rinsed with IPA for 20 min and cured in a convection oven for 2 h at 80 °C. Finally, all tridimensional nanocomposites obtained were stored at room temperature and protected from light before all characterizations. 

**Characterization:** X-ray diffraction (XRD) analyzes were performed using a PANalytical Empyrean X-ray diffractometer at 2θ intervals from 5 to 80°, with a step size of 0.03°, for 0.3 s, with Cu Kα radiation of 1.540598 Å. The morphological characterization and elemental mapping were conducted by scanning electron microscopy (SEM) using a JEOL JSM-6490LV microscope with an acceleration voltage of 20 kV, by placing the samples on copper tape. The surface composition of the coating was analyzed by high-resolution X-ray photoelectron spectroscopy (HR-XPS) using a Thermo Scientific Escalab 250 Xi with step energy of 200 eV for survey scans and 50 eV for high-resolution scans, with step sizes of 1.0 and 0.1 eV, respectively, in a high-purity argon atmosphere. The selected region spectra covered the elements C1s, O1s, and Ti2p. The nanocomposites were analyzed by attenuated total reflectance (ATR) using a Frontier MIR PerkinElmer ATR-Fourier-transform infrared (FTIR) spectrometer at 4000–400 cm^−1^, using 12 scans and 0.4 cm^−1^ resolution at room temperature. Inductively coupled plasma (ICP) spectroscopy was carried out to quantify the amount of titanium present in each sample, using a high-performance ICP emission spectrometer (Thermo Scientific iCAP 6500 ICP-OES). Thermal stability was studied by thermogravimetric analysis (TGA), which was performed using a TA Instruments SDT Q600 system with alumina crucibles and heating from room temperature to 900 °C at 10 °C/min under a feed of ultrahigh-purity N_2(g)_. Differential scanning calorimetry (DSC) analyses were conducted using a TA Instruments DSC Q200; 1–2 mg of each sample was placed into an aluminum pan, and all analyses was performed under a N_2(g)_ (99.99%) flux of 100 mL/min. Each sample was analyzed as follows: (i) heating at 10 °C/min from 0 °C up to 200 °C; (ii) isothermal at 200 °C for 1 min; (iii) cooling at 10 °C/min from 200 °C down to 0 °C; and (iv) heating at 10 °C/min up to 210 °C. Thermal transitions were obtained from the second heating. Tensile and compressive forces were tested utilizing a universal testing machine (model AGX plus) using a load cell of 100 kN and a crosshead speed of 5 and 1.3 m/min, respectively, according to ASTM D695 [1]. Six samples of each system were tested for these studies, and the average results were reported. Tribological test performance was studied under dry sliding conditions using a ball-on-disc tribometer (Anton Paar) under ambient conditions (relative humidity of 30−40%), at a load of 5 N and a sliding speed of 0.05 m/s, with 100 Cr6 balls of 6 mm in diameter as counterparts, according to ASTM-G99. 

## 3. Results and Discussion

A straightforward way to achieve strong and lightweight 3D-printed polymer nanocomposites by SLA based on resin/TiO_2_ NPs (R_TiO_2_ X wt.%) was disclosed for the first time, as shown in Figure 1a. Here, different amounts of TiO_2_ NPs were used (X = 0.0, 0.2, 0.4, 0.8, or 1.2 wt.%) and dispersed in 3 mL of IPA to assure a homogeneous suspension, which was added to 130 g of resin and mixed by mechanical stirring. Six ASTM D638 Type IV solid tensile specimens for each dispersion were 3D printed at 45°; meanwhile, six solid cylindrical structures (ASTM D695) for compression test specimens, along with six hollow structures (gyroid type) and six complex structures (dense lattices), were 3D printed in a horizontal orientation, following a CAD. 

The presence of TiO_2_ NPs was confirmed in all 3D-printed polymer nanocomposites, following the diffraction intensity at Bragg angles of 2θ = 25.6° and 27.5°. These angles correspond to the diffraction from the (101) and (110) planes of the anatase and rutile phases according to ICDD cards 04-001-7641 and 01-073-6031, respectively, as shown in Figure 1b. Thus, the presence of TiO_2_ NPs (P25) within the grey resin was confirmed. Indeed, as the weight percentage of TiO_2_ NPs increased, the diffraction peaks became more intense and noticeable. Additionally, the crystallite sizes of the TiO_2_ NPs were calculated from the full width at half-maximum at the values of 2θ = 25.6° and 27.5° using the Scherrer equation at 25.3 and 45.4 nm for the anatase and rutile phases, respectively, in good agreement with previous reports in the literature [18,19,20]. High-resolution X-ray photoelectron spectroscopy (HR-XPS) revealed typical binding energies around 458.8 eV (Ti 2P_3/2_) and 464.5 eV (Ti 2P_1/2_) [21], corresponding to two representative 3D-printed polymer nanocomposites at 0.8 and 1.2 wt.% TiO_2_ NPs (See Figure 1c). Moreover, the difference in binding energy between the two peaks was 5.7 eV due to the spin–orbital coupling. This value suggests that these two peaks correspond to Ti^4+^ in the TiO_2_ nanostructure [22].

Inductively coupled plasma (ICP) spectroscopy was used to quantitatively confirm the amounts of TiO_2_ NPs present in 3D-printed samples, which were consistent with the amounts of TiO_2_ NPs added to the resin, as shown in Table 1. 

Specifically, the printability states and surface finishing of the hollow samples (bicontinuous structures), dense lattices, and solid cylindrical structures at several TiO_2_ NP contents can be observed in Figure 1d. Interestingly, the addition of TiO_2_ NPs did not affect the reproducibility of the printing process or the resolution of self-supported hollow or solid structures (cylinders)—even at 1.2 wt.% nanoparticles. Nonetheless, in dense structures (i.e., many tinny elements bonded together), a slight decay in the printability was observed at high contents of nanoparticles.

On the other hand, the effect of TiO_2_ NP contents on the thermal properties of all of the 3D-printed samples was characterized by the glass transition temperature (T_g_), the crystallization temperature (T_c_), and the melting temperature (T_m_) via differential scanning calorimetry, as shown in Figure 2a–c. Figure 2a shows the T_g_ of the 3D-printed polymer nanocomposites. It is very noticeable that with a slight increase in TiO_2_ NP contents, the T_g_ increases as well. The 3D-printed control sample, without any added nanoparticles, had a T_g_ of 81.5 °C. It was observed that the T_g_ increased to 82, 82.1, 82.3, and 82.7 °C when increasing the nanoparticle contents by 0.2, 0.4, 0.8, and 12 wt.%, respectively. These results suggest that the presence of TiO_2_ NPs does not affect the chain mobility of the polymer resin, but they have an essential effect on the mechanical properties, as mentioned below.

The T_c_ for each nanocomposite (see Figure 2b) was obtained during the first cooling step, showing that there were no significant differences in the T_c_ of the printed and cured samples, as demonstrated by the slight decrease in the T_c_ (around 0.6–0.9 °C) with the increase in the amount of TiO_2_ NPs. These results suggest that the presence of TiO_2_ NPs does not have any relevant influence on the photocurable 3D printing process in the interleaved layer. Interestingly, it is important to highlight the shape of the thermogram during the cooling process, showing a more remarkable “slope” in the control sample from T_c_ to 50 °C. This slope faded with the increase in the nanoparticle content, becoming a flat line at 1.2 wt.% TiO_2_ NPs. This behavior could be strongly related to the heat dissipation phenomenon after the exothermic T_c_. In the control sample, the heat absorbed by the cured resin during the heating cycle was released more slowly during the cooling step, compared with faster heat dissipation or no residual heat from the 3D-printed nanocomposites. In other words, the TiO_2_ NPs play a remarkable role as effective heat absorbers. The T_m_ values for all of the polymer nanocomposites are shown in Figure 2c, allowing us to explain the opposite phenomenon to the previous one observed for T_c_. The T_m_ trends slightly decreased with the addition of 0.4 wt.% TiO_2_ NPs, and showed a tendency to increase with 0.8 and 1.2 wt.% contents. This behavior could be attributed to the homogeneous dispersion of the TiO_2_ NPs in the resin, as demonstrated by SEM–EDS in the following subsection.

At low contents of TiO_2_ NPs, the dispersions were very homogeneous, without the presence of aggregates (≤0.4 wt.%), while at high contents, the TiO_2_ NPs started to form aggregates (≥0.8 wt.%). Bearing these effects in mind, we can highlight two main effects: (i) at low contents of TiO_2_ NPs, there is no heat diffusion from the nanoparticles to the matrix; hence, the heat absorbed by the resin is effectively only related to the heat necessary to melt the crystalline phase (lamellae) from the polymer matrix; (ii) at high contents of TiO_2_ NPs, the agglomerations play the role of hotspots, absorbing the initial heat; consequently, the system needs more heat to reach the T_m_ and, thus, the T_m_ tends to increase. We believe that this increase in the T_m_ could be more evident if a higher content of nanoparticles in the nanocomposite (>5 wt.%) is achieved.

Furthermore, the ATR-FTIR spectra corresponding to all the 3D-printed polymer nanocomposites are shown in Figure 2d. Vibrational stretching modes of C-H bonds were observed and attributed to methyl and methylene groups at 2900 cm^−1^—specifically, *ν_as_* CH_3_ at 2953 cm^−1^, *ν_s_* CH_3_ at 2860 cm^−1^, *ν_as_* CH_2_ at 2917 cm^−1^, and *ν_s_* CH_2_ at 2850 cm^−1^. The latter is consistent with vibrations at 1450 cm^−1^, corresponding to the bending mode of the C-H bond, *δ_s_* CH_2_. In addition, *δ_s_* CH_3_ was observed at 1370 cm^−1^. On the other hand, a broad signal was observed at 3400 cm^−1^, attributed to the N-H bond from a secondary amine. Likewise, the vibration signal for the double bond of the carbonyl functional group (C=O) was at 1700 cm^−1^, present in identified compounds derived from methacrylate, polyesters, and polyurethanes. In this sense, the vibrations corresponding to C = O and N-H bonds suggest the presence of amides or urethane groups in the commercial resin. In this case, the low amount of TiO_2_ NPs is imperceptible; their presence does not affect the bonding of the chemical species during the vat polymerization process.

On the other hand, compressive and tensile analyses were performed to study the contribution of the low loadings of TiO_2_ NPs to the mechanical properties of the 3D-printed polymer nanocomposites. Six 3D-printed solid cylinders (height = 2 cm, diameter = 1 cm) corresponding to each formulation were tested (using the ASTM D695 compression test) to study their compression resistance properties. Figure 3 shows the compressive stress (σ), compressive strain (ε), yield strength, and toughness data as a function of the TiO_2_ NP contents for all of the 3D-printed polymer nanocomposites, along with the control sample for comparisons.

Interestingly, the polymer nanocomposites exhibited an outstanding compression resistance at 0.4 wt.% TiO_2_ NPs, supporting 5 tons before breaking—representing an increase of around 200%. Furthermore, all properties increased simultaneously at this loading, which is not trivial. After 0.4 wt.% of nanomaterial, the compressive strain remained constant, but the toughness, the compressive stress, and the yield strength decreased as a function of the TiO_2_ NP content. Thus, 0.4 wt.% TiO_2_ NPs seems suitable for obtaining a homogeneous dispersion in the resin that produces reinforcement in each layer. Between 0 and 0.4 wt.% nanomaterial content, an improvement in the compressive properties was observed. This is because their addition can fill the matrix’s pores, reduce its porosity and rugosity, increase its relative density, and improve its mechanical properties, as other authors have reported [23,24]. However, beyond 0.4 wt.%, the concentration of nanoparticles increases as well as their clustering, producing agglomerations and fracture spots. In this sense, all properties decay drastically except for the compression strain, which remains unalterable.

The tensile stress–strain (σ–ε) curves for all polymer nanocomposites are shown in Figure 4a. All of the 3D-printed tensile bars presented a minimal plastic deformation (ε < 5%), suggesting that the added TiO_2_ NPs are incorporated between the acrylate polymerized layers that make up the print. The more NPs are added, the greater their contribution to fragility. The mechanical properties (σ, ε, and Young’s modulus) as a function of the wt.% of TiO_2_ NPs added to the nanocomposites showed a decrease once the TiO_2_ NPs were added to the resin, in concordance with the compressive test results. 

It is important to mention that using a high concentration of nanoparticles or microparticles higher than the printed layer’s height can produce stress concentration centers in the matrix, leading to brittle materials [24,25]. In our case, we used nanoparticles homogeneously distributed on the polymer matrix, as further demonstrated by SEM–EDS. In addition, UTS (ultimate tensile strength) was practically reduced by around 14%, indicating that the nanofiller can alter the structure of the polymeric resin, affecting the curing kinetics (by reducing the crosslinking reaction during the bath polymerization) and promoting a negative impact on the mechanical properties of the cured resin [26,27]. Figure 4b shows an optical image of the 3D-printed samples.

In all cases, a homogeneous dispersion was achieved, observed as a change in tonality from dark gray towards light gray as the amount of TiO_2_ NPs increased, as shown in Figure 4b. It is worth noting that adding a large amount of nanofillers to the photopolymerizable resin produces a laser scattering, leading to a poor print quality, as was observed in sample R_TiO_2_ 1.2 wt.%. This phenomenon was observed by other authors previously [23]. SEM images of TiO_2_ NPs (0.0, 0.2, 0.4, 0.8, and 1.2 wt.%) after the tensile tests are presented in Figure 4c–g, where the imperfections on the surface progressively disappear as the TiO_2_ NP content increases. SEM imaging was also used to examine the morphology of TiO_2_ NPs in the solid polymer matrix after 3D printing. Figure 4e shows an image of irregular resin platelets formed after the dog–bone tensile bars break. In this case, when the amount of TiO_2_ NPs increases, the homogeneous distribution of the nanoparticles favors the appearance of the breaking point in the layers of the photopolymerized resin (Figure 4c–g). 

Energy-dispersive X-ray spectroscopy (EDS) was performed on representative samples to analyze the chemical composition of the 3D-printed objects. SEM–EDS showed a uniform Ti distribution, as observed in Figure 4h–j. According to the results, an apparent increase in the Ti content in the matrix is observed when the additive percentage increases, revealing the chemical composition to be 0–0.7% titanium. Furthermore, a uniform distribution of titanium can be observed on the surface. Nonetheless, samples with the highest TiO_2_ NP contents showed nanoparticle clustering, leading to brittle materials.

On the other hand, the tribological behaviors of 3D-printed polymer nanocomposites at a normal load of 1 N during 4000 sliding cycles are shown in Figure 5a. The friction coefficient (µ) of R_TiO_2_ 0.4 wt.% gradually became stable, exhibiting the lowest friction coefficient obtained (µ = 0.2). Interestingly, the rest of the polymer nanocomposites, including the control sample (neat resin), exhibited different tribological behaviors. In most cases, µ progressively increased or showed a slightly erratic pathway, as in the sample R_TiO_2_ 0.8 wt.%. Figure 5b–f show the wear tracks obtained for all of the 3D-printed materials, which were wider (~450 µm) for polymer composites with unstable µ. Thus, R_TiO_2_ 0.4 wt.% (Figure 5d) exhibited a wear track ~300 µm wide.

## 4. Conclusions

This work demonstrates the ability to produce strong, high-toughness, and lightweight polymer nanocomposites with low friction coefficients and complex geometries via 3D printing and SLA using low loadings of nanoparticles (TiO_2_ < 1 wt.%). Several tridimensional structures—including solid tensile bars, solid compressive test specimens, hollow gyroid-type structures, and dense hollow lattices—were straightforwardly 3D printed using an affordable and commercial desktop 3D printer. We obtained 3D-printed objects with outstanding compressive mechanical properties, able to support up to 5 tons of weight before breaking, 0.4 wt.% of TiO_2_ NPs. Furthermore, these polymer nanocomposites exhibited a low friction coefficient of ~0.2. Finally, we believe that 3D printing of ceramic composites is a powerful tool to achieve strong and lightweight polymer nanocomposites, which are highly desirable for the aerospace and automotive industries.

## Data Availability

Not applicable.

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
