# Peer review of "Strong and Lightweight Stereolithographically 3D-Printed Polymer Nanocomposites with Low Friction and High Toughness"

_polymers, 2022, doi:10.3390/polym14173628_

Round 1
Reviewer 1 Report
In the present work, the authors have proposed a strategy that allows making strong and lightweight 3D polymer nanocomposites with a low-frictional coefficient and high toughness prepared by stereolithography for the lower loading of TIO2. It’s a novel work and meaningful for the manufacturing of 3D printed polymer nanocomposites. However, the following problems should be addressed before acceptance for publication.
How about the bonding strength when sealing the open structure with the lightweight 3D printed polymer nanocomposites?
Is there any relationship between the particle size of TiO2 and mechanical properties of the composites? What happens if using TiO2 with different size?
To provide good background about the polymer nanocomposites, authors may cite the following references, 1) DOI: 10.1002/mame.201600553 2) DOI: https://doi.org/10.1007/s40964-021-00232-z 2)DOI: 10.1002/pat.3818
How about the mechanical strength of the 3D-printed polymer after the sintering of the nanocomposite? Have the authors compared it with other nanocomposites?
In the transfer of the microstructure, would the 3D polymer microstructure deform? How significant is the deformation?
Could the authors comment on why the compressive strain remains constant, but the toughness, the compressive stress, and the yield strength decrease as a function of TiO2 content.? Is this due to Tg effects? Other effects?
Author Response
We sincerely appreciate all valuable and thoughtful Reviewer comments, which helped us improve our contribution's quality. Our responses to Reviewer 1 comments are described below in a point-to-point manner. Appropriated changes suggested by the Reviewer have been introduced to the manuscript. "Please see the attachment"

Reviewer 2 Report
Comments to authors:
In this manuscript, the authors have reported the fabrication of strong and lightweight 3D printed nanocomposites with low friction and toughness. Although 3D printing with TiO2 has been widely reported, this work provides some useful information about the effect of additives on the friction and toughness of materials. Thus, a minor revision is recommended. Some suggestions or comments are listed as follows.
1. The structure of V4 should be provided in the manuscript.
2. The discussion of the results was not convincing. For example, why does TiO2 addition negatively influence the mechanical properties of composites (stress-strain)? An SEM-EDS should be helpful in analyzing the distribution of TiO2 in the composites and offer useful information to the results.
3. I believe that the synthesized nanocomposites were strong rather than tough since the tensile strain was less than 10%. Moreover, how about its performance (mechanical properties) as compared to other 3D printing V4 materials?
Author Response

(The authors gave the same response as above.)
